# Risk-Sensitive Markov Decision Processes of USV Trajectory Planning with Time-Limited Budget

**DOI:** 10.3390/s23187846

**Published:** 2023-09-13

**Authors:** Yi Ding, Hongyang Zhu

**Affiliations:** 1Maritime College, Guangdong Ocean University, Zhanjiang 524091, China; dingyi@gdou.edu.cn; 2College of Mathematics and Computer, Guangdong Ocean University, Zhanjiang 524091, China

**Keywords:** trajectory planning strategy, trajectory optimization, Risk-Sensitive Markov decision processes

## Abstract

Trajectory planning plays a crucial role in ensuring the safe navigation of ships, as it involves complex decision making influenced by various factors. This paper presents a heuristic algorithm, named the Markov decision process Heuristic Algorithm (MHA), for time-optimized avoidance of Unmanned Surface Vehicles (USVs) based on a Risk-Sensitive Markov decision process model. The proposed method utilizes the Risk-Sensitive Markov decision process model to generate a set of states within the USV collision avoidance search space. These states are determined based on the reachable locations and directions considering the time cost associated with the set of actions. By incorporating an enhanced reward function and a constraint time-dependent cost function, the USV can effectively plan practical motion paths that align with its actual time constraints. Experimental results demonstrate that the MHA algorithm enables decision makers to evaluate the trade-off between the budget and the probability of achieving the goal within the given budget. Moreover, the local stochastic optimization criterion assists the agent in selecting collision avoidance paths without significantly increasing the risk of collision.

## 1. Introduction

Ship trajectory planning is an imperative task for navigation safety at sea. The goal of trajectory planning is to calculate a safe and comfortable trajectory for the unmanned agent to accomplish the intended driving task [1]. An Unmanned Surface Vehicle (USV) is a light intelligent surface-carrying unmanned piece of equipment with a small size, low cost, fast speed, and strong maneuverability [2]. Collision-free path generation is one of the core technologies of an USV’s autonomous navigation capability [3]. An USV’s collision avoidance algorithm can be considered a local path-planning algorithm. Some scholars have tried the design of a ship navigation safety domain to solve the ship collision problem [4,5,6]. However, most of them only consider static obstacles or semi-dynamic obstacles that do not change in the course [7,8]. A highly ideal motion model is used in collision avoidance [9]. Some collision avoidance studies ignored balance efficiency and effectiveness [10]. Some other researchers have evaluated the generated collision avoidance trajectories from the perspective of risk assessment [11], using multiple parameters such as navigation risk [12], navigation smoothness [13], and other metrics. In [14,15], the authors discussed ASV(Autonomous Surface Vehicle) developments in depth, while conflict detection and obstacle avoidance received a lesser amount of attention. Only a few studies related to reacting collision avoidance for unmanned ships were included in the paper [16]. Traditional obstacle avoidance algorithms mainly study the smoothness and safety of the trajectory without considering the time constraints [17], and the usual obstacle avoidance algorithms require iteration through all possible states [18], which greatly increases the computational complexity of the algorithms [19].

Recently, planning-based methods have been widely applied in various fields [20], including artificial intelligence [21], robotics [22], and operations research [23,24]. These methods are especially applicable in complex and dynamic environments where coordination of multiple actions is required to achieve the goal [25]. In the trajectory planning of Unmanned Surface Vehicles (USVs), improving image quality is crucial for effective navigation. Particularly in challenging environments such as sea fog, the visual system of USVs may be compromised. To address this issue, several studies have explored image-defogging techniques for nighttime conditions, aiming to enhance the image clarity of USV visual systems [26,27]. By integrating these techniques into USV trajectory planning, navigation performance in complex environments can be improved, ensuring safe and efficient operations. There are various methods of path planning available, such as path planning based on genetic algorithms, fuzzy logic, and optimization algorithms. For instance, path planning methods that use genetic algorithms can locate optimal routes in intricate environments by simulating biological evolution processes, such as an enhanced ant colony optimization algorithm for designing search and rescue routes [28,29]. However, this method may require extensive computational resources and time for optimization. In contrast, fuzzy logic-based path-planning techniques employ fuzzy reasoning and online learning, such as a WIP vehicle control approach based on an enhanced artificial potential field for multi-obstacle environments [30]. This method can adapt to changes and uncertainties in the environment. However, the method could be more arduous in modeling the environment’s geometric features and dynamical constraints [31]. Furthermore, there exist other path-planning methods such as the optimization-based RRT-A* path-planning method [32] which could require more effort in obtaining and modeling obstacle information of the environment and may have restricted adaptability to intricate and dynamic obstacle environments. However, the objective function of the USV trajectory planning problem is very complex and involves the processing of a large amount of different information [33]. Due to a large number of constraints and mathematical modeling difficulties, a number of problems remain to be solved in existing trajectory planning systems [34]. These include the handling of trajectory constraints [35], real-time performance [36], and the lack of trajectory substitution [37]. Inspired by the design of reliable systems, Yu et al. [38] proposed a class of planning problems called Risk Sensitive Markov Decision Processes (RS-MDP), where the goal is to find a policy that maximizes the probability of cumulative costs without exceeding a user-defined budget [39].

The USV travel process has distinct states, and each state corresponds to the same action space [40]. Thus, the process of optimizing an USV’s trajectory planning depends on the existing state, prediction of possible future collisions, optimization judgment, and the subsequent state selection based on the prediction. The idea is that the USV reaches its destination without collision and without exceeding the formulated time budget. The traveling process of the USV can be divided into different states, and the cost consumed for each state transition depends on the current state of motion of the USV and the time budget. The decision-making process stops irreversibly when the time required to generate a local budget reaches a certain point when the path trajectory exceeds expectations, or when collision avoidance is unsuccessful before reaching the destination [41]. In this state, no matter what action is taken, it results in a local path search failure, which is called a “dead end” for an USV. Therefore, this paper aims to determine a set of measures to prevent the USV from reaching a “dead end” between the start and goal states. The objective is to maximize the probability that the total cost (time consumption) of local path planning for the USV remains within a specified time constraint [42].

This paper’s principal contributions are based on the aforementioned concepts.

This paper proposes a standard method for probabilistic planning to optimize the trajectory planning of an USVs considering uncertain time costs. The approach utilizes a goal-oriented Markov decision process to find strategies that minimize the expected cumulative cost from the initial state to reach the goal.In this study, it is assumed that the decision maker has a predetermined upper budget limit. However, we seek to investigate whether a lower budget would imply an acceptable probability of the optimal cost threshold in the policy, given the risk attitude. Therefore, the objective is to find the best balance between the danger of a situation leading to a dead end and the initial budget.Trajectory optimization planning for USVs is accomplished by transferring states between action and transition probabilities. The optimization of USV trajectories is achieved by moving states between action and transfer probabilities. The problem being investigated consists of two phases. In the design phase, the number of time intervals and the potential states that can be produced within those intervals are determined. In the second phase, the USV is programmed to move from the start state to the goal state in a manner that maximizes the probability of the cumulative cost of local path planning for the USV being within a specific duration. To achieve this, a series of velocity angles and yaw angles must be determined.

This paper consists of the following sections: Section 2 presents the dynamic model of an USV based on the Risk-Sensitive Markov decision process (RS-MDP). Section 3 describes in detail the generation and evaluation of the optimal trajectory strategy with time constraints. Section 4 presents the results of computational experiments conducted to evaluate the proposed algorithm. Based on simulation experiments, a comparison was made between the effects of three different algorithms on the selection of a collision avoidance path for an USV, and the degree of excellence resulting from various collision avoidance factors in the path selection was analyzed. Section 5 contains the discussion of conclusions.

## 2. Problem Statement and Risk-Sensitive Markov Decision Process (RS-MDP)

### 2.1. Problem Description

This section describes a USV trajectory optimization planning problem with an uncertain time cost. The series consists of a set of $s=\left\{s_{0},s_{1},\ldots ,s_{t}\right\}$ of USV position states, which performs a set of $Action=\left\{A_{0},A_{1},\ldots ,A_{t}\right\}$. The actions have a time cost and at the end of each action time, the USV moves to the next state. Each action produces a series of different next states, depending on the current state of the USV and the action to be chosen. In other words, the task processing time changes as the USVs select different actions.

The aim of this paper is to produce an execution strategy for USVs that maximize the probability of successfully completing the given task before the deadline. This implies generating trajectories with movement times shorter than the current time limit.

### 2.2. RS Markov Decision Process (RS-MDP) for USV

This section presents how the MDP can be applied to solve this problem. We consider the problem of planning a temporal constraint USV trajectory. Given a 2D static map of the environment representing occupied and free space, and a goal state, trajectory *S*. The trajectory *S* is a continuous function, and a goal-oriented Risk-Sensitive Markov Decision Process (RS-MDP) is defined as a tuple < S, $S_{0}$, *A*, *P*, *C*, *G*, $T_{\delta }$ >, where $S_{0}$ is a finite state set; *A* is a finite action set; *P* is a transition function that returns the probability $P({\tilde{S}}_{i}$, $a^{i}$, ${\tilde{S}}_{i+1}$) of reaching state ${\tilde{S}}_{i+1}$ from state ${\tilde{S}}_{i}$ when the USV applies action $a^{i}\in A$; *C* is a cost function that associates a cost $C(S_{i},a^{i})$ when action $a^{i}\in A$ is executed in state ${\tilde{S}}_{i}\in \tilde{S}$; *G* is a set of goal states; and $T_{\delta }$ is the user-defined limited resources(e.g., time) associated with the trajectory, which in this paper refers to the time spent traveling the trajectory. We assume that the goal state is an absorbing state with zero cost and that non-goal absorbing states have positive costs.

#### 2.2.1. A Set of Possible States of RS-MDP

In this section, we begin by formalizing the class of MDP problems in which we are interested. With the circle group *T*, the trajectory *S* is a continuous function that maps time to the USV position and the heading(yaw) angle ψ. For certain applications, such as mapping and surveillance, one may opt to define a sequence with m−1 intermediate waypoints. We denote $\tilde{S}$ = [${\tilde{S}}_{0}$, ${\tilde{S}}_{1}$, …, ${\tilde{S}}_{i}$, …${\tilde{S}}_{m-1}$, ${\tilde{S}}^{end}$] to be attained in order. ${\tilde{S}}_{0}\in \tilde{S}$ is the initial state, ${\tilde{S}}^{end}\in \tilde{S}$ is the final state. We denote $s_{r}\left(i\right)$ as the position of the USV, v(i) as the instantaneous velocity of the USV at its current position $s_{r}\left(i\right)$, and $s_{\psi }\left(i\right)$ as the heading angle ψ of the USV at state S(i).
(1)$$\left\{\begin{array}{c} S\left(i\right)={\left[{s_{r}\left(i\right)}^{T}\;s_{\psi }\left(i\right),v\left(i\right)\right]}^{T}\\ {s_{r}\left(i\right)}^{T}={\left[S_{x}\left(i\right)\;\;S_{y}\left(i\right)\right]}^{T} \end{array}\right.$$

#### 2.2.2. Action Function of RS-MDP

An action *A* is a set of all possible actions. A(s) defines the set of actions that can be taken in state *S*. An action $a^{i}=<△v^{i},△\psi^{i}>$ defines the deterministic transition between states $s_{r}\left(i\right)\overset{a^{i}}{\to }s_{r}\left(i+1\right)$, and $a^{i}\in A$. The conversion of the USV angle between two adjacent states is given in detail in Figure 1. $S_{\psi }^{i}$ is the USV heading angle at waypoint $S_{i}$. Figure 1 shows the process of state $s_{r}\left(i\right)$ to state $s_{r}\left(i+1\right)$ is the trajectory of the USV from waypoint $S_{i}$ to waypoint $S_{i+1}$, ∡μ represents the minimum angle at which the USV must turn to avoid a collision, which is considered as a threshold velocity of collision avoidance. ∡μ can be obtained by the Velocity Obstacle model [43].

$△\psi^{i}$ is the USV heading angle turned from state $s_{r}\left(i\right)$ to state $s_{r}\left(i+1\right)$. We define the following constraints in the angle of a state on a feasible trajectory:(2)$$\left\{\begin{array}{c} △\psi^{i}=∠S_{\psi }^{i+1}-∠S_{\psi }^{i}\\ ∡\mu \leq △\psi^{i} \end{array}\right.$$

The trajectory generation is described by three degrees of freedom, where (x,y,ψ) is the state of the waypoint $S_{i}$. An action $a^{i}=<v^{i},△\psi^{i}>$ defines the deterministic transition between states $s_{r}\left(i\right)\overset{a^{i}}{\to }s_{r}\left(i+1\right)$, and $v^{i}\in \left[0,v_{max}^{i}\right]$, $v_{max}^{i}$ is the maximum velocity the USV can reach from state $S_{i}$, and the action $a^{i}$ represents the change of motion of the USV, calculated as Equation (Equation 3), where $\nu^{i}\in a^{i}$ is the velocity taken by the USV at state $s_{r}\left(i\right)$.
(3)$$\begin{array}{c} S_{x}\left(i+1\right)=S_{x}\left(i\right)+\nu^{i}\cos\psi \\ S_{y}\left(i+1\right)=S_{y}\left(i\right)+\nu^{i}\sin\psi \end{array}$$
Typically, since there are many actions available on state $S^{i}$. In order to measure the extent achieved by the USV after performing a certain action, we summarize five types of actions, according to the rules of collision avoidance, which are: $A_{1}\left(△v^{i}>0,△\psi^{i}=0\right)$ describes the acceleration of the USV in the current direction; $A_{2}\left(△v^{i}<0,△\psi^{i}=0\right)$ characterizes the deceleration of the USV in the current direction; $A_{3}\left(△v^{i}<0,△\psi^{i}>0\right)$ depicts the USV deceleration and steering to avoid collisions; $A_{4}\left(△v^{i}=0,△\psi^{i}>0\right)$ describes the USV maintaining constant speed and steering to avoid collisions; and $A_{5}\left(△v^{i}=0,△\psi^{i}=0\right)$ gives a description of the USV traveling in the current direction, at a constant speed. The USV starts to move from the initial state and stops gently at the end of the time or path, while the acceleration/deceleration of the USV is limited to safe navigation. According to the International Regulations for Preventing Collisions at Sea (COLREGs), ships should choose to pass from the right side (i.e., their own left side) of an oncoming ship [44]. Therefore, all heading angle changes defined in this paper are greater than or equal to zero.

#### 2.2.3. The Cost Function of RS-MDP Search Strategies

Typically, since there are many actions available on state $S^{i}$. In order to measure the extent achieved by the USV after performing a certain action, we use a utility function to evaluate it. In this paper, combining the core idea of USV trajectory planning: USVs always drive to the position with a high safety factor and close to the destination, which is consistent with the stochastic Boltzmann policy [45] in the thermodynamic principle: molecules all tend to move to the position with low energy consumption. Therefore, in order to predict alternative paths of the USV to the destination and to allow deviations from the optimal policy, this algorithm will combine a stochastic Boltzmann policy to relax the constraint and obtain the best matching way of the actions, i.e., the optimal policy, denoted as $\psi_{g}$, which assigns a probability to each action a to be executed in state *s*, as shown in Equations (Equation 4) and (Equation 5). The USV in state *i* finds the successor state node $S_{i+1}$ through the probability *p* in Equation (Equation 4). $C(a^{i})$ is constructed as a weighted sum of the Euclidean distance covered by *a*, and the unitary cost of the target state C(s), provided by the optional input semantic map.
(4)$$\begin{array}{c} p\left(S^{i+1}|S^{i},a\right)\propto e^{-C(s^{i},a)/K} \end{array}$$
(5)$$\begin{array}{c} C\left(s_{r}^{i},a^{i},s_{r}^{i+1}\right)=C\left(S_{r}^{i+1}\right)+C\left(S_{r}^{i+1}|a^{i},S_{r}^{i}\right) \end{array}$$
States transfer probability, as shown in Equation (Equation 6).The probability of a history $h_{\pi }$ is given by Equation (Equation 7):(6)$$P^{\pi }\left(s^{\prime }|s\right)=\sum_{a\in A}\pi \left(a|s\right)P\left(s^{\prime }|s,a\right)$$
(7)$$P\left(h_{\pi }\right)=\prod_{t=0}^{\infty }P^{\pi }\left(s^{\prime }|s\right)$$

A policy for an RS-MDP is a function π, and $\pi \left(s^{t},t\right):S\times A⟼\left[0,1\right]$ is the rule for selecting actions at time *t*, which specifies which action $a^{i}$ to take for each state $s_{r}\left(i\right)$ and time step. A history $h_{\pi }$ = (${\tilde{S}}^{start}$, ${\tilde{S}}_{1}$, …, ${\tilde{S}}_{i}$, … ) for the policy π is a valid sequence of states obtained by executing policy π starting from state $S^{start}$. The historical cumulative cost on trajectory ${\tilde{S}}^{i}$ is denoted as $C(h_{\pi })$, note that policy π is allowed to pick an action *a* in state s that exceeds the current available time. In Equation (Equation 8), we denote *t*=$\left[t_{1},\ldots ,t_{m}\right]$ as the time allocation over trajectory segments, e.g., $t_{j}$ is the time allocated for the segment between states or waypoints (note that we see waypoint *i* and the state $S_{i}$ as the same) $S_{i-1}$ and $S_{i}$. When planning trajectories through obstacle-rich environments, it may be preferable to incorporate collision avoidance constraints instead of states. For each *t* in the definition domain $\left(\left[0,t_{max}\right]\right)$, there are states *s* that satisfy the following conditions: kinematic constraints on the USV, collision constraints, and physical limitations of the USV.
(8)$$\sum_{j=1}^{i}t_{j}=T^{i},i=1,2,\ldots ,m-1$$
The time constraint for the transition between successor states is given by Equation (Equation 8).The cumulative cost of history $h_{\pi }$ is given by:(9)$$\left\{\begin{array}{c} C\left(h_{\pi }\right)=\sum_{i=0}^{\infty }C\left(s^{t},\pi \left(s^{t},t\right)\right)\\ C\left(s^{t},a^{t}\right)>0\\ a^{t}\in \pi \left(s^{t},t\right) \end{array}\right.$$

The deterministic transition between states s in the Markov reward process. The strategy π satisfies the following equation. The probability of taking action *a*:(10)$$P\left(h_{\pi }\right)=\prod_{t=0}^{\infty }P\left(s^{t},\pi \left(s^{t},t\right),s^{t+1}\right)$$
The set of all histories of policy π that ends in a goal state is denoted as $H_{\pi }$. From Equations (Equation 7)–(Equation 10), we obtain the probability of policy π generating a history that starts at state s and reaches a goal state without exceeding a user-defined voyage time $T_{\delta }$.
(11)$$P^{\pi }\left(s,\delta \right)=\sum_{h_{\pi }\in H_{\pi },s^{0}=s,c\left(h_{\pi }\right)\leq T_{\delta }}P\left(h_{\pi }\right)$$
Alternatively, the above probability is called the cost-threshold probability for policy π from states *s* and the restricted time $T_{\delta }$. When no histories satisfy the time constraint, the probability is zero. The optimum cost-threshold probability is then defined as follows for a given state *s* and constraint time $T_{\delta }$:(12)$$P^{*}\left(s,T_{\delta }\right)=\max_{\pi }P^{\pi }\left(s,T_{\delta }\right)$$

Figure 2 shows an example of a RS-MDP. Figure 2a illustrates the state transition diagram for an example of a RS-MDP with four states. It displays potential transitions between states as well as the corresponding actions that can be executed. Nodes in the diagram represent states, and they are labeled as $s_{0},s_{1},s_{2},$ and $s_{3}$. The arrows that link the nodes represent the state transitions, displaying the different possible routes that the system can take. Each transition has an action and cost linked to it. To illustrate, we can examine the transition between states $s_{0}$ and $s_{1}$. This transition is represented by the arrow labeled as ‘action $a_{1}$’ as shown in Figure 2b. This transition indicates that the system can progress from state $s_{0}$ to state $s_{1}$ by taking the action $a_{1}$, which incurs a specific cost.

In Figure 2b, $S=(s_{0},s_{1},s_{2},s_{3})$, $G=(s_{3})$, $s_{0}$ is the initial state, $A=(a_{1},a_{2})$, the transition function is depicted as labeled arcs, the cost function C(s,a) annotates outgoing arcs and $T_{\delta }=2$. The different policies are due to the uncertainty of the cost function and time prediction. For instance, we can obtain some polices such as: $\left\{s_{0},a_{1},s_{1},a_{2},s_{3}\right\}$, $\left\{s_{0},a_{2},s_{2},a_{2},s_{3}\right\}$, $\left\{s_{0},a_{1},s_{1},a_{1},s_{2},a_{2},s_{3}\right\}$. Actions $a_{i}$ are represented by squares and transitions $P(s,a,s^{\prime })$ are represented by arcs. As the constraint time grows, the number of arcs for an augmented state increases. Since more budget is available, more actions can be applied.

The RS-MDP transitions the action space by adjusting the control inputs of the unmanned vessel. Adjusting the acceleration and heading angle changes the motion behavior of the unmanned vessel, thus controlling the increase or decrease of its speed. Acceleration adjustments allow the USV to increase or decrease its speed, or even come to a complete stop. The heading angle is the angle of the USV relative to the reference direction. Adjusting the heading angle can change the direction of the USV’s sailing path. Modifying the heading angle allows the USV to move in a different direction. By adjusting these control inputs, the USV can perform a variety of actions, transition between different states, and successfully achieve goals. For example, in a given state, increasing the acceleration and modifying the steering angle can cause the USV to move towards a goal state. Upon reaching the target state, the USV can be stabilized by appropriately decreasing the acceleration and modifying the steering angle.

## 3. Trajectory Planning Based on MHA Algorithm

Trajectory planning involves moving from point A to point B while avoiding collisions over time. This can be calculated using both discrete and continuous methods. In essence, trajectory planning includes path planning as well as planning how to move based on velocity, time, and kinematics. For the path requirements, both velocity and acceleration are continuous. Here, ${\tilde{S}}^{start}$ and ${\tilde{S}}^{end}$ represent the start and end points, respectively, and each consists of a prescribed position and heading angle. The motion trajectory points of the USV can be determined from the state collection by considering the heading of the current node and the position and heading of the selected waypoint (next node). Figure 3 illustrates the motion planning flow of the USV operating in free space.

The distribution shows that states with low time costs of consumption always have a higher chance of being occupied.
$$\begin{array}{c} C\left(S^{i},a\right)=\left(\lambda +1\right)\left(\alpha -1\right) \end{array}$$
where
(13)$$\begin{array}{c} \alpha =\left\{\begin{array}{cc} 1 & if\;\mathrm{dist}(S_{x,y}^{i+1},goal)=0\\ \infty & if\;\mathrm{dist}(S_{x,y}^{i+1},obstacle)\leq R\\ \frac{\mathrm{dist}(S_{x,y}^{i+1},goal)}{\mathrm{dist}(S_{x,y}^{i+1},obstacle)} & else \end{array}\right. \end{array}$$
Equation (Equation 13) shows that $\mathrm{dist}\left(\psi_{0},\psi_{t}\right)=\sqrt{{\left(x_{t}-x_{0}\right)}^{2}+{\left(y_{t}-y_{0}\right)}^{2}}$, the value is the Euclidean distance between the two points. When $\mathrm{dist}(S_{x,y}^{i+1},goal)=0$, it means that the USV has already reached the destination, and at this time no cost is consumed to arrive with probability 1. λ∈[0,1] are the obstacle densities in the $S^{i+1}$ state, *R* is the anti-collision threshold of the USV for the integration of the speed barrier algorithm with the ship safety domain [46]. If $\mathrm{dist}(S_{x,y}^{i+1},obstacle)\leq R$, it means that the USV has collided with an obstacle in that state. At this time the cost is infinite and cannot be chosen.
(14)$$C\left(S^{i+1}|a,S^{i}\right)=C\left(a\right)+C\left(S^{i}\right)$$

### 3.1. RS-MDP for Initialized Cost Threshold Probabilities

Unlike a standard goal-oriented MDP, the optimal policy for the RS-MDP is not stationary, which prevents the direct application of standard MDP algorithms. Seeking to improve the convergence of dynamic programming for the RS-MDP, we develop more efficient solutions for the RS-MDP under the local stochastic optimization criterion in this study. Equation (Equation 15) denotes the probability of transition to state $s^{\prime }$ after the execution of action *a* in state *s*, while θ represents the time budget cost set by the user. In path-planning problems, θ acts to restrict the amount of time expended on path planning, thereby limiting the path-planning algorithm to identifying the most effective route within a predetermined time frame. This probability is identified as $P(s,\theta ,a,s^{\prime })$ and can be assessed through the examination of past data or by utilizing the system model. In a path-planning problem, this probability may be represented as the probability of moving to state $s^{\prime }$ following the selection of action *a* in state *s*.
$$P^{*}\left(s,\theta \right)=\max_{a\in A}\sum_{s^{\prime }\in S}P\left(s,\theta ,a,s^{\prime }\right)$$
where:(15)$$P\left(s,\theta ,a,s^{\prime }\right)=\left\{\begin{array}{cc} 0 & \;\mathrm{if}\;C(s,a)>\theta ,\\ 1 & \;\mathrm{if}\;C\left(s,a\right)\leq \theta \;\mathrm{and}\;s^{\prime }\in G,\\ P\left(s,a,s^{\prime }\right)P^{*}\left(s^{\prime },\theta -C\left(s,a\right)\right) & \;\mathrm{if}\;C\left(s,a\right)\leq \theta \;\mathrm{and}\;s^{\prime }\notin G. \end{array}\right.$$
Local Stochastic Optimization is able to plan increasingly fast trajectories as the model improves by classifying candidate trajectories as either feasible or infeasible.The optimal policy for every budget level that is equal to or smaller than the constrained time $T_{\delta }$ is defined by
(16)$$\begin{array}{cc} \; & \pi_{T}^{*}\left(s\right)\in \arg\max_{\pi }P^{\pi }\left(s_{0},T\right),\;\;\mathrm{for}\;\mathrm{each}\;T\in \left[0,T_{\delta }\right]\\ \; & \\ \; & s.t.\left\{\begin{array}{c} S\left(0\right)={\tilde{S}}^{start}\\ S\left(T\right)={\tilde{S}}^{end}\\ S\in \xi_{path}\cap \xi_{all}\\ \sum_{}T\leq T_{\delta }\\ \dot{S(T)}=0 \end{array}\right. \end{array}$$
(17)$$P^{*}\left(s,T\right)=P^{*}\left(s,\;T_{j}\right),\;\mathrm{for}\;\;T_{j-1}\leq T<T_{j}$$
Equations (Equation 15)–(Equation 17) illustrate the relationship between the probabilistic models in trajectory planning. Equation (Equation 15) evaluates historical probability given a policy, Equation (Equation 16) computes the probability of the optimal path given a time restriction, and Equation (Equation 17) estimates the overall optimal probability of the trajectory by computing the probability of the optimal path across distinct time intervals. Together, these equations form the computational process of probabilistic modeling in trajectory planning.
(18)$$r\left(s,a,s^{\prime }\right)=\left\{\begin{array}{cc} -1 & \mathrm{if}\;C(s,a)>\theta \\ \frac{P\left(s,\theta ,a,s^{\prime }\right)}{C(S^{\prime }|a,S)} & \mathrm{if}\;C\left(s,a\right)\leq \theta \;\mathrm{and}\;s^{\prime }\in G,\\ -\frac{P\left(s,\theta ,a,s^{\prime }\right)}{C(S^{\prime }|a,S)} & \mathrm{if}\;C\left(s,a\right)\leq \theta \;\mathrm{and}\;s^{\prime }\notin G. \end{array}\right.$$

In Equation (Equation 18), $r(s,a,s^{\prime })$ represents the reward value based on the given conditions. $\frac{P\left(s,\theta ,a,s^{\prime }\right)}{C(s^{\prime }|a,s)}$ is the ratio of the accessibility of state *s* to state $s^{\prime }$ to its cost. It can be used to measure the efficiency of travelling from state *s* to state $s^{\prime }$, which is the probability of reaching state s for a unit cost. The higher the ratio, the more efficient it is to arrive in state *s* from state $s^{\prime }$. The function $r(s,a,s^{\prime })$ returns a reward value based on this ratio, as well as other conditions. Specifically, if the cost of action *a* from state *s* is greater than threshold θ, the reward value is set to −1. If the cost is less than or equal to θ and the resulting state $s^{\prime }$ is in the goal set *G*, the reward value is set to $\frac{P\left(s,\theta ,a,s^{\prime }\right)}{C(S^{\prime }|a,S)}$. On the other hand, if the resulting state *s* is not in the goal set *G*, the reward value is set to $-\frac{P\left(s,\theta ,a,s^{\prime }\right)}{C(S^{\prime }|a,S)}$.

Algorithm 1 describes how the choice of the probability distribution of the initial cost threshold affects the search space and the convergence performance of the algorithm. The effectiveness and performance of the optimization algorithm can be improved by choosing the probability distribution of the initial cost threshold appropriately.
**Algorithm 1** Initialized cost threshold probabilities**Input:** Initial state *s*, goal state *g*, time limit *T***Output:** Optimal path $p^{*}$ to reach *g*  1:Initialization  2:Set *s* as the current location of the USV  3:Set *g* as the destination of the USV  4:**for** t = 1 to T **do**  5:      **for** *a* in possible actions **do**  6:            Observe next state set $s^{\prime }$ and reward $r(s,a,s^{\prime })$  7:            Add experience $\left(s,a,r,s^{\prime }\right)$ to experience replay buffer  8:      **end for**  9:    Update cumulative cost $C(s^{t},\pi \left(s^{t},t\right))$ and policy function π(s) using experience replay buffer and RS-MDP algorithm10:     Select action *a* with lowest C(s, a)-value from policy function π(s) for current state *s*11:       Move USV to next state $s^{\prime }$ and receive reward $r(s,a,s^{\prime })$12:       **if** thens=g13:             Break out of the loop14:       **end if**15:**end for**16:Return optimal path $p^{*}$ to reach *g*

### 3.2. Adaptive Threshold Search Algorithm for RS-MDP

We propose a version of the classical dynamic programming solution for the MDP, called the Adaptive Threshold Search algorithm, which is based on the concept of LOS. This algorithm is an improved version of LOS for the RS-MDP in the augmented state space, which addresses the finite-policy MDP problem with reinforcement learning. By generating only augmented states that belong to histories that end in a goal state and by verifying probability convergence at a given budget at each iteration, we demonstrate how LOS efficiency can be improved.

#### 3.2.1. Value Iteration for RS-MDP

Since the optimal cost threshold probability is the immovable point of the Bellman equation for the RS-MDP (Equation (Equation 16)), we can compute it by iteratively applying the following Bellman update rule (estimated cost threshold probability at iteration *i*) until (near) convergence, starting from an acceptable probability. An overview of the algorithm is shown in Algorithm 2. The main parameters involved are as follows: ϵ is error tolerance. It represents the allowed range of error in the final result. The algorithm stops iterating when the difference between P(s,θ) and $P^{*}\left(s,\theta \right)$ is less than ϵ. The algorithm searches and calculates the state transition probabilities, continuously updating the expected time of the best path for each state and time combination. It decomposes the problem into a series of state transitions and action choices, selecting the action with the $val_{max}$ (highest “val” value) as the best path’s expected time for the current state. Whether there exists a policy with a cumulative reward less than or equal to the upper bound threshold.
**Algorithm 2** Adaptive Threshold Search algorithm (ϵ,θ)  1:**Input:**  2:         ϵ: a small positive value representing the desired accuracy  3:         θ: a parameter used for threshold calculation  4:**Output:**  5:         True if a condition is met, False otherwise  6:**for** all ((s,θ)∈T) **do**  7:      $P^{*}\left(s,\theta \right)=0$  8:**end for**  9:$c_{max}:=\max_{s,a\in A}C\left(s,a\right)$10:**if** $CHECKP(\theta -1,c_{max})>0$**then**11:      **return** ture12:**end if**13:**for** all ((s,θ)∈T) **do**14:      $val_{max}=0$15:**end for**   16:**for** all a∈A **do**17:      val=018:**end for**19:**for** each $T\in T^{\theta }$ **do**20:      **repeat**21:      i:=022:      θ:=023:      **for** all ((s,θ)∈T) **do**24:             $P^{i+1}\left(s,\theta \right):=\;\max_{a\in A}\sum_{s^{\prime }\in S}\left\{\begin{array}{cc} 0 & \;\mathrm{if}\;C\left(s,a\right)>\theta \\ P\left(s,a,s^{\prime }\right)*\mathrm{SEARCH}P\left(s^{\prime },\theta -C\left(s,a\right)\right) & \;\mathrm{if}\;\left(s^{\prime },\theta -C\left(s,a\right)\right)\notin T,C\left(s,a\right)\leq \theta \\ P\left(s,a,s^{\prime }\right)*P^{i}\left(s^{\prime },\theta -C\left(s,a\right)\right) & \;\mathrm{if}\;\left(s^{\prime },\theta -C\left(s,a\right)\right)\in T,C\left(s,a\right)\leq \theta \end{array}\right.$25:             **if** $\left|P^{i+1}\left(s,\theta \right)-P^{i}\left(s,\theta \right)\right|>\delta$ **then**26:                  $val=val+T\left(s,a,s^{\prime }\right)\cdot P\left(s^{\prime },\theta -C\left(s,a\right)\right)$27:                  $\delta :=\left|P^{i+1}\left(s,\theta \right)-P^{i}\left(s,\theta \right)\right|$28:            **end if**29:            i:=i+130:      **end for**31:      **until** δ<ϵ32:      **for** all (s,θ)∈T **do**33:            $P^{*}\left(s,\theta \right):=P^{i}\left(s,\theta \right)$34:      **end for**35:**end for**36:**return** false

#### 3.2.2. Cost-Threshold Probability Convergence

Algorithm 3 verifies that the probability model in the USV path-planning problem satisfies the collision avoidance risk and cost consumption preferences. It ensures the consistency of the probabilistic model by comparing the values of the probability transfer function that are computed at two different thresholds. Algorithm 4 sets thresholds by searching for distinct combinations of thresholds and states to locate the probability transfer function that meets specific requirements. It determines the threshold value at which the probability transfer function of all states complies with the specified condition. This process assists in selecting the appropriate threshold value to ensure the optimal performance of the path-planning model in diverse states.
**Algorithm 3** CHECK P(s,θ)1:**if** $\theta <c_{\max}$**then**2:      return false3:**end if**4:**for** all s∈S **do**5:      **if** $P\left(s,\theta \right)\neq P\left(s,\theta -c_{\max}\right)$ **then**6:           return false7:      **end if**8:**end for**

**Algorithm 4** SEARCH P(s,θ)
1:**for** all $\left(s^{\prime },\theta^{\prime }\right)$ in decreasing sort by $\theta^{\prime }$ **do**2:      **if** $P\left(s,\theta \right)\neq P\left(s,\theta -c_{\max}\right)$ **then**3:            return false4:      **end if**5:
**end for**



## 4. Simulations and Discussion

Two algorithms, dynamic window approach with virtual (DWV) [17] and State Lattice Planner (SLP) [45], are considered in this study, where the DWV and SLP are considered conventional methods, and the MHA is envisioned as a proposed improved method. The simulations will show the quality of the operation of such an algorithm. This paper observed a collision-avoiding simulation using a minimum safety domain $d_{safe}$ for vessels. Moreover, the influence of navigational behaviors and environmental impacts (wind and currents) is ignored in the modeling process. We set ϵ=0.01 as the residual error, and all the algorithms in this study run on MATLAB 2019a.

Table 1 shows the initial parameters of the USV. The optimal offset of the USV heading angle is calculated under the condition of constant speed, the expansion of the barrier is set to 0.8 m.

### 4.1. Optimization of Trajectory
Planning with Limited Budgets

As shown in Figure 4a, the random map size is 210×210 (grid size), the blue point is the initial position of the USV, and the red point is the goal. The start and goal positions are (USVxstart, USVystart) = (30.0, 15.0) and (GBxgoal, GBygoal) = (180.0, 200.0).

Figure 4b illustrates the gradient colors used to represent the velocity magnitude of the USV at different positions. This visual representation provides an intuitive illustration of how the USV velocity changes in the given path. Faster speeds are indicated by lighter colors in the bit-shift representation, whereas darker colors indicate slower speeds. This coding method swiftly obtains the USV’s velocity change throughout the designated path. The correlation between colors and speeds is exploited to precisely identify likely speed bottlenecks or congested areas, enabling the assessment and optimization of path-planning models.

According to the initial settings of parameters in Table 1, parameters are shown in Table 2, which displays simulation results in different time budgets, including the success rate, trajectory length (TL), and the average travel time. The algorithm was tested in 50 iterations where successful planning had to be completed within 100 ms. We set ϵ=0.01 as the residual error.

Table 2 provides a correspondence between the path length and the time budget to show how the path changes with different time budgets, offering guidance for path-planning decisions in practical applications. Table 2 presents the metrics connected with generating trajectories, such as trajectory length, time, and cost. State node utilization is a measure of the planning algorithm’s efficiency, calculated by dividing the algorithm utilization by the planned nodes and total sampled nodes. The performance of the algorithm is evaluated using the algorithm duration and success rate. Reducing the time budget significantly lowered the node sampling rate to 90.84% in the test results. In addition, frequent computational timeouts led to a reduced success rate of 90%. Moreover, the algorithm processing time was longer (89.06 ms).

Figure 5 illustrates two different paths based on two different time budgets. Figure 5a displays the path map under various time budgets. Color-coded path lines clearly reveal the optimal path selection of the USV under various time constraints. The path length does not differ significantly under a time budget of 45 s. This implies that the USV produces optimum paths at high speeds by selecting the state node with the smallest steering angle as often as possible, guaranteeing safe navigation and collision avoidance while reaching the target location. The path becomes flatter with a longer time budget of 65 s. This enables the USV to explore the area for a more extended period and choose flatter routes to gather additional information or complete intricate tasks.

The gradient colors of the displacement trajectory curves in the displacement transfer representation graph in Figure 5b represent the USV’s velocity magnitude at different points, illustrating how the USV’s speed varies along the path. Slower speeds are represented by lighter colors, and faster speeds are indicated by darker colors in the displacement transfer representation graph. As the time budget decreases, there is a higher likelihood of the USV selecting the maximum speed to traverse the path. However, the path length does not reduce correspondingly, given the low probability of the USV changing its heading angle, which is set for collision avoidance risk preference.

### 4.2. Comparative Algorithm Simulation Experiments

Figure 6 shows the evolution of four moving features during simulation in the time budget for 65 s. Figure 6a demonstrates that a change in the Course Angle parameter of the DWV has a minimal effect on path planning. The text suggests that the DWV prioritizes the stability of the heading over the curvature of the path. The Course Angle parameter has a greater impact on path planning in the SLP. The statement suggests that the SLP prioritizes path curvature to increase its flexibility in adapting to changes in the environment. When comparing the first two algorithms, it was found that in the MHA, the variation of the Course Angle parameter has a moderate effect on path planning. The statement suggests that the MHA achieves a balance between heading stability and path curvature.

Figure 6b shows that the DWV significantly affects path planning when the Heading Angle parameter varies. This indicates that the DWV is highly sensitive to the Heading Angle and fluctuates more when the Heading Angle varies. The SLP, in contrast to the DWV, has a lower impact on path planning when the Heading Angle parameter changes. This suggests that the SLP prioritizes maintaining heading stability in the path trajectories over fine-tuning the heading angle, which may result in less success within the predetermined time budget. The MHA demonstrates a moderate effect on path planning when the Heading Angle parameter changes. This highlights that the MHA achieves a balance between adjusting the heading angle and maintaining heading stability.

By combining Figure 6c,d, we can observe that in the DWV, higher translational acceleration and velocity parameters lead to a faster acceleration of the USV, resulting in faster arrival at the target position. This is useful in situations that require rapid arrival. The SLP prioritizes smooth acceleration and deceleration processes with less emphasis on the changes to translational acceleration and velocity parameters during path planning. This is useful in situations where smoothness is a priority. The MHA considers both speed and stability, balancing the impact of changes in translational acceleration and velocity parameters during path planning. This results in moderate effects on path planning, ensuring both speed and smoothness. This approach is useful in situations where both speed and stability are important.

It can be observed in Table 3 that the MHA algorithm outperforms other algorithms with its high success rate and low average angular velocity integral under different time budgets. The SLP algorithm performs better in terms of the average angular velocity integral but is relatively poor in terms of success rate and average time. The DWV algorithm has poor performance in terms of average path length. On the other hand, the MHA algorithm has the highest success rate under all-time budgets. This can be attributed to its use of state-space transfer probabilities in path planning, which optimizes the sampling of state nodes, thereby enhancing the effectiveness of the search space. Trajectory evaluation is a simultaneous consideration of multiple objectives such as shortest path, obstacle avoidance, and minimum angular velocity integral. Considering these objectives together, the MHA algorithm can find better paths, which in turn increases the success rate. The SLP algorithm’s relatively low success rate can be attributed to its use of a simplified single-objective optimization approach in path planning. Since the SLP algorithm considers only a single objective (e.g., shortest path or obstacle avoidance) without comprehensively analyzing other factors, it is unable to find optimal paths in complex scenarios and thereby leads to a reduced success rate.

The average time of the MHA algorithm under the three time budget requirements is 34.00 s, 59.772 s, and 76.278 s, respectively, which all satisfy the requirement that the trajectory sailing time does not exceed the time budget. This is because the MHA algorithm can dynamically adjust the weight allocation of the state node selection according to the different scenarios and requirements. This can balance the trade-offs between different goals and find the appropriate path efficiently.

In all available time budgets, the DWV algorithm has the longest average path length. On the other hand, the MHA algorithm more accurately predicts future states and, as a result of its ability to estimate the transfer probability of USVs taking different actions, plans paths more efficiently. Achieving the above requires precise modeling of state transfer and elimination of unnecessary path nodes. The SLP algorithm has a high average angular velocity integral of 2.194 deg/s in all available time budgets. Conversely, the MHA algorithm combines its objectives in path planning: it determines the shortest path and sets a minimum steering angle threshold. This implementation results in a lower average angular velocity integral. Consequently, the MHA algorithm identifies paths that are both quick and comfortable, thereby lowering the angular velocity integral.

## 5. Conclusions

This study uses a Risk-Sensitive Markov decision process model to propose a method for generating a collection of states within the search space of collision avoidance for USVs. It plans the movement routes using an improved reward and cost function. The results of the simulation experiments are analyzed, and it is found that the MHA algorithm allows decision makers to make trade-offs between budget and objectives when planning trajectories within budget. The local stochastic optimization criterion assists USVs in selecting collision avoidance paths without significantly increasing risk. The MHA algorithm outperforms in complex environments. In complex environments where many obstacles and constraints such as narrow passages and different types of obstacles are present, the MHA algorithm takes these factors into account and finds the optimal path that meets the time budget while adapting to the collision avoidance needs. However, the methodology is subject to limitations and constraints. Firstly, additional research is required to determine the optimal cost threshold probabilities under a low budget in order to achieve the best balance between budget and objectives. Secondly, our approach may have limitations in addressing uncertainty, particularly in complicated and dynamic environments. Additionally, we have neglected to consider the influence of navigation behaviors and environmental factors on the model, which necessitates further investigation into their incorporation. Lastly, our study requires validation through field experiments to evaluate its performance and feasibility in practical settings. Therefore, future research could overcome these limitations and enhance the applicability and precision of the approach.

## Figures and Tables

**Figure 1 sensors-23-07846-f001:**
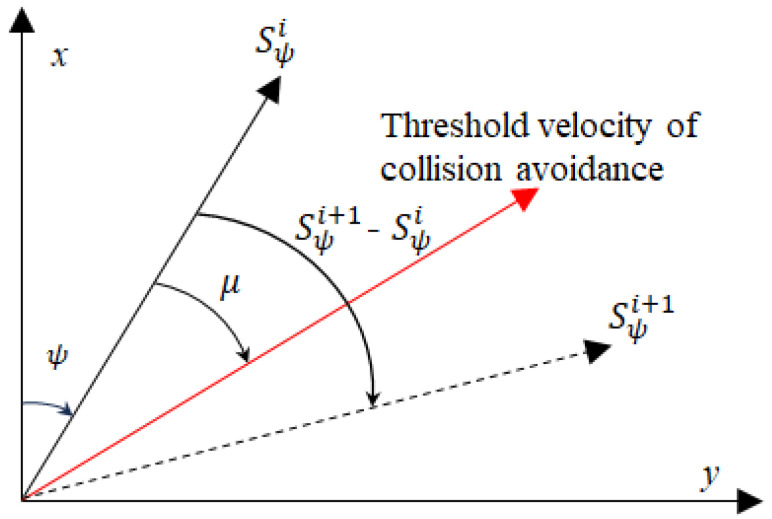
States under heading angle ψ change.

**Figure 2 sensors-23-07846-f002:**
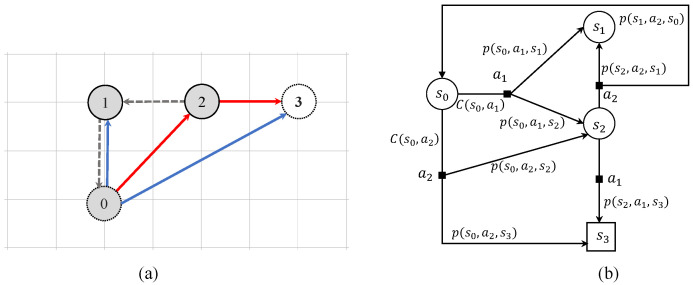
Example of RS-MDP with four states and integer costs: (**a**) state transition diagrams; (**b**) action variation diagrams associated with time costs.

**Figure 3 sensors-23-07846-f003:**
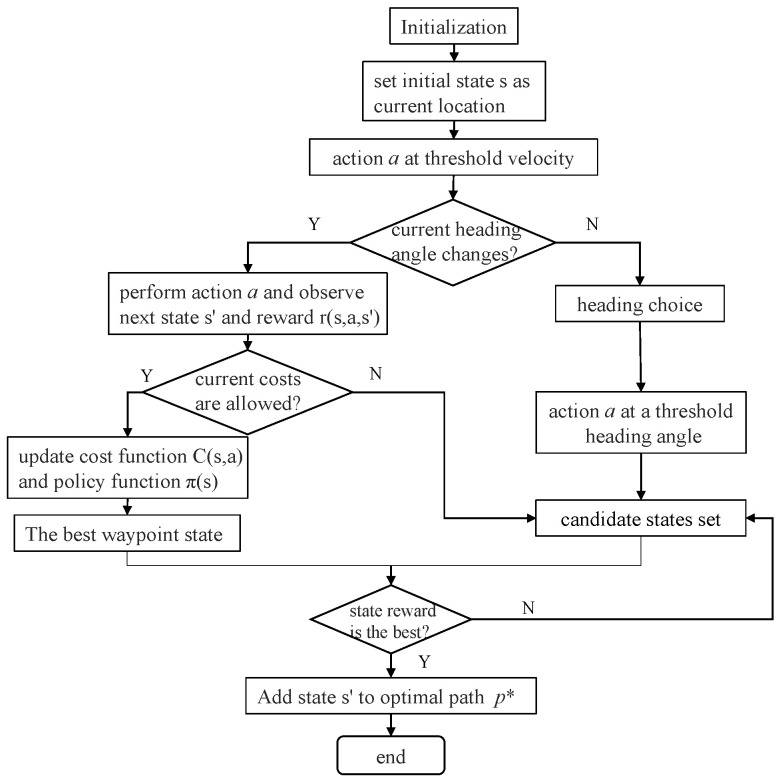
Motion planning flow of the USV operating.

**Figure 4 sensors-23-07846-f004:**
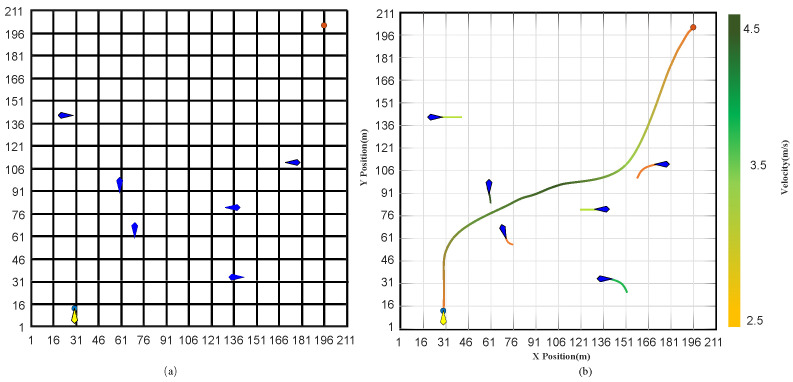
Analysis of USV path planning: random map, initial goal positions, and velocity distribution. (**a**) Random map, initial goal positions. (**b**) Visualization of velocity distribution.

**Figure 5 sensors-23-07846-f005:**
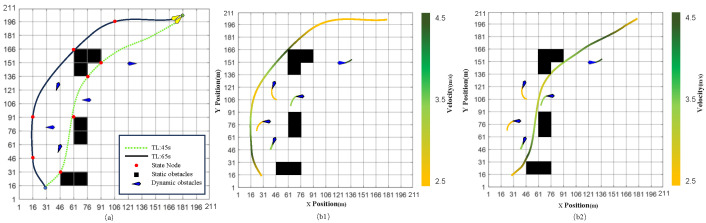
Comparison of a path generated in different time budgets: (**a**) path map under various time budgets. (**b1**) Time budget for 65 s. (**b2**) Time budget for 45 s.

**Figure 6 sensors-23-07846-f006:**
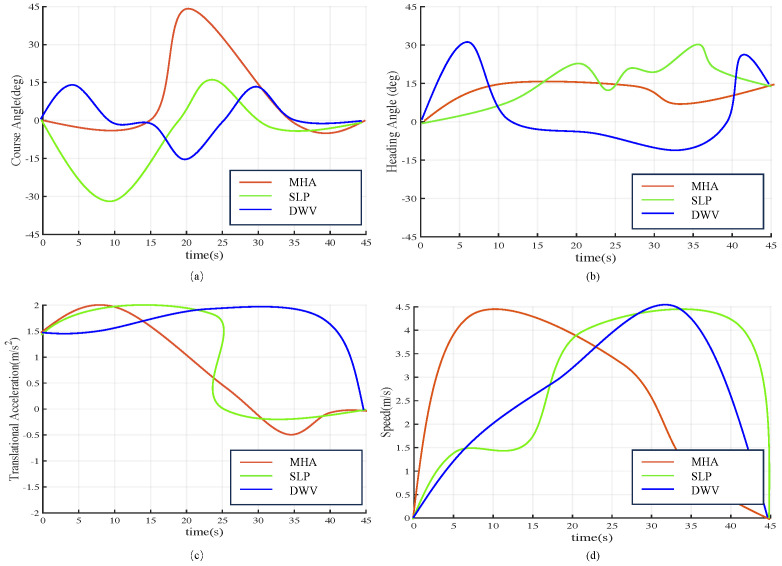
Three algorithms simulation parameters compared in time budget of 65 s: (**a**) Course angle. (**b**) Heading Angle. (**c**) Translational Acceleration $\left(m/s^{2}\right)$. (**d**) USV speed.

**Table 1 sensors-23-07846-t001:** Constraint initial parameter values.

Parameters	USV	Target Vessel
Maximum Translational Velocity	4.5 [m/s]	6 [m/s]
The average obstacle gap	−0.3 [m/s]	0.0 [m/s]
Maximum Angular Range of Heading Angle	45 [deg]	0 [deg]
Minimum Angular Range of Heading Angle	−45 [deg]	0 [deg]
Maximum Angular Velocity	5 [m/s]	-
Minimum Angular Velocity	−5 [m/s]	-
Time Step	0.1 [s]	0.1 [s]
Maximum Predicted Time	5 [s]	-
Maximum Translational Acceleration	2 [$m/s^{2}$]	0 [$m/s^{2}$]

**Table 2 sensors-23-07846-t002:** Simulation results in different time budgets.

Time Budget	State Node Utilization	Success Rate [%]	Time [s]	Trajectory Length [m]	Algo. Time [ms]
65	97.84	96	62.03	199.521	80.21
45	90.84	90	43.79	195.804	89.06

**Table 3 sensors-23-07846-t003:** Compare algorithms at different time budgets.

Time Budget	Method	Succ. [%]	T.vel [s]	TL [m]	Mean.vel [$m/s^{2}$]	Ang.vel. Integral [deg/s]
	DWV	82.340	34.57	154.093	4.881	2.340
35	SLP	12.36	49.76	92.077	3.094	1.823
	MHA	80.231	34.00	105.060	3.090	1.340
	DWV	12.340	81.653	270.843	3.317	1.800
60	SLP	34.554	88.910	358.574	4.033	2.560
	MHA	88.231	59.772	182.902	3.060	3.230
	DWV	91.630	76.157	249.871	3.281	2.101
80	SLP	93.800	82.976	173.752	2.094	3.471
	MHA	97.781	76.278	238.140	3.122	0.822

## Data Availability

Not applicable.

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
