# Peer review of "Risk-Sensitive Markov Decision Processes of USV Trajectory Planning with Time-Limited Budget"

_sensors, 2023, doi:10.3390/s23187846_

Round 1
Reviewer 1 Report
This paper has proposed a heuristic algorithm with time-optimized USV avoidance based on a risk-sensitive Markov decision process model (MHA algorithm). This method uses a risk-sensitive Markov decision process model to generate a set of states in the USV collision avoidance search space, which is formed based on the locations and directions that can be reached by the current set of states matched by the USV with the time cost of the set of actions. Combining with an improved reward function and the (constraint time-dependent) cost function, the USV can plan practical motion paths that match its actual time cost. Extensive experiments demonstrate that the proposed MHA algorithm allows the decision maker to observe the trade-off between the budget and the probability of reaching the goal within the budget, while the local stochastic optimization criterion helps the agent to better select the collision avoidance path without significantly increasing the collision avoidance risk. The detailed comments are as follows:
1. The theoretical foundation of this paper is robust.
2. The primary contributions of this submitted manuscript lack clarity for readers. I suggest that the authors present these contributions in a bullet-point format.
3. Within the introduction section, the authors have reviewed ship trajectory planning algorithms. While these methods adequately address normal circumstances for USV trajectory planning, the performance of the proposed MHA algorithm under nighttime and hazy conditions remains unclear. It is advisable for the authors to explore recent nighttime and haze algorithms [1-2] to ascertain potential solutions for USV trajectory planning in such challenging scenarios.
[1] Nighttime defogging using high-low frequency decomposition and grayscale-color network
[2] Multi-purpose Oriented Single Nighttime Image Haze Removal Based on Unified Variational Retinex Model
4. The references cited may be outdated. I recommend that the authors review and include more recent articles published between 2022-2023.
5. The comparisons provided in Table 3 are insufficient to substantiate the superiority of the proposed MHA algorithm. Could you expand the selection of methods beyond DWV and SLP for a more comprehensive comparison?
6. Could you discuss the limitations of the proposed MHA algorithm?
It appears that the quality of English language in the manuscript could benefit from some improvement. There are instances where sentence structures and phrasing could be refined for greater clarity and coherence. Attention to grammar, punctuation, and word choice could enhance the overall readability and professionalism of the paper.
Reviewer 2 Report
The authors proposed an MDP-based trajectory planning. Why the research problem is interesting and well defined, there are some parts that I believe can be improved:
- There are some parameters and functions in the paper which are not defined. For example, \theta in Figure 1 is not defined, later \theta is used in eq (15) and beyond which again is not clear what is it. Q(s,a) in Fig 3 is not defined, the value function V(s) in Alg. 1, and so on.
- Some functions are used with different sets of arguments and formats. For example, cost function. eq. (14) is a good example that shows the cost function is used with various input format.
- what is C_{max}? in Alg. 2 it is used in line 4 but its value is assigned in line 16
- Captions are very short and not informative. Add some info to captions for figures, tables, and algorithms. (e.g Alg. 3 and Alg. 4 or Fig. 4).
- Captions in section 4.1 should show that the results are for proposed algorithm
- Fig4.a and Fig. 4.b can have the same scale.
- Fig. 5 caption should be improved. (a) is for both budgets and b1 and b2 are not clear that belong to which time budget
- Why OCAP method in other authors' paper is not compared with the proposed algorithm?
- Some references used the word proceeding multiple times. I understand this may come from the original source but it should be manually fixed. for example, references 29 and 30 shows multiple proceeding or the year 1999
- There are some minor writing mistakes. "unmanned Surface Vehicle (USV)" can be "Unmanned Surface Vehicle (USV)", many of in text citations need a space before such as "effectiveness[10]", what is ASV? a typo or not defined abbreviation "authors discussed ASV"
- Equations should be punctuated as far as i know. please try to use fewer paramaeters/math lettrs if possible or define them properly. I have past part of MDPI policy here:
"Punctuate equations as part of a regular sentence. For example, if the equation comes at the end of a sentence, a period should be placed immediately after the equation. It is not necessary to always use a colon to end the paragraph before an equation. If the equation is followed by “where . . . ” to define the symbols used, “where” should be all lower case and flushed to the margin (without first line indentation) to indicate that it does not begin a new paragraph. All terms used in an equation should be defined in the text. It is highly recommended to check specifically for this during proofreading before submission, as undefined terms could lead reviewers and editors to misinterpret your meaning. Additionally, be aware of multiply defined symbols, and we recommend using standard notation in the field where it exists (e.g., P for a probability function). The format (italics/non-italics) of each character in an Equation should be consistent with the main text."
- what does MHA stand for?
discussed above.
Round 2
Reviewer 2 Report
Thanks for addressing the comments. It seems most of the issues are solved. However, there can be more improvements:
- For captions, assume a reader asks you what this figure/algorithm/table and you are going to answer it in one or two sentences. This can be a good caption. Some of them are not informative yet.
- Equations are not punctuated properly yet. I will leave this to the editor and MDPI to decide.
- Please use the full term of MHA right before the first use in the abstract.
-My main concern is still the confusing use of Q-value, rewards and value iteration. Alg 1. Step 10 points to select the best action based on Q-value which seems unrelated to this work! Also, the reward function that is used in this algorithm and in Figure 3 is not defined anywhere in the paper. This reduces paper reproducibility by other researchers.
Round 3
Reviewer 2 Report
I have no further comment
Author Response
Dear reviewer:
Thank you for reviewing my manuscript. I appreciate your feedback. Thank you for your time and consideration.